# Concerns of Home Isolating COVID-19 Patients While Receiving Care via Telemedicine during the Pandemic in the Northern Thailand: A Qualitative Study on Text Messaging

**DOI:** 10.3390/ijerph19116591

**Published:** 2022-05-28

**Authors:** Kanokporn Pinyopornpanish, Nopakoon Nantsupawat, Nida Buawangpong, Suphawita Pliannuom, Tanat Vaniyapong, Wichuda Jiraporncharoen

**Affiliations:** 1Department of Family Medicine, Faculty of Medicine, Chiang Mai University, Chiang Mai 50200, Thailand; kpinyopo@gmail.com (K.P.); nidalooknum@gmail.com (N.B.); suphawita.pliannuom@gmail.com (S.P.); wichudaj131@gmail.com (W.J.); 2Department of Surgery, Faculty of Medicine, Chiang Mai University, Chiang Mai 50200, Thailand; tanat.v@cmu.ac.th

**Keywords:** telemedicine, telehealth, teleconsultation, COVID-19, home isolation, text message

## Abstract

As there were strict limits on contact between health professionals and patients during the COVID-19 pandemic, telemedicine increased in importance with regard to improving the provision of health care and became the preferred method of care. This study aims to determine the topics of concern expressed by individuals with COVID-19 receiving care at home via teleconsultation. The qualitative study was conducted using secondary data of chat messages from 213 COVID-19 patients who had consented to online consultation with the health care team. The messages were sent during the home isolation period, which was between 29th October and 20th December 2021. Thematic analysis was used to analyze the data. All patients had consented to the use of their data. A small majority of the patients were female (58.69%). The average age was 32.26 ± 16.92 years. A total of 475 questions were generated by 150 patients during the isolation period. Nearly thirty percent (29.58%) never asked any questions. From the analysis, the questions could be divided into three themes including: (1) complex care system; (2) uncertainty about self-care and treatment plan with regard to lack of knowledges and skills; and (3) concern about recovery and returning to the community after COVID-19 infection. In conclusion, there were enquiries about many aspects of medical care during home isolation, detailed answers from professionals were useful for the self-care of patients and to provide guidance for their future health behavior. The importance of the service being user friendly and accessible to all became increasingly evident.

## 1. Introduction

Since the end of 2019, the world has been suffering from the coronavirus disease (COVID-19) pandemic [1]. The exponential rise in people infected with COVID-19 has been increasing the challenges for healthcare professionals globally. This included Thailand, where the total number of infected people reached 2.1 million, with the number of deaths totaling over twenty thousand in 2021 alone [2,3]. Health care facilities were key to overcoming this situation. Due to the accelerating number of patients, the number of beds available in the hospital became limited, so as a result, home isolation programs were established. In October 2021, the home isolation program was launched, which is similar to the policy invoked in many countries for example Turkey [4], Germany [5], and India [6]. This service allows people with COVID-19 to receive treatment at home and be monitored by the incorporation of telemedicine into the care system. The aims of the system included the detection of life-threatening conditions such as respiratory failure and sepsis [7].

As there were strict limits on contact between the health professionals and patients during the COVID-19 pandemic, telemedicine increased in importance with regard to improving the provision of health care and became the preferred method of care. As suggested by a recent systematic review [8], the use of teleconsultation helps to provide a service nearly equivalent to real-time consultation, and reduce the risk of physical contact and exposure to infected patients. This also allows a smaller team to remotely manage patient concerns and prevent morbidity, while minimizing the risks of transmission from in-person contact to healthcare workers and other patients [9,10,11,12].

Thailand, a middle-income country, has been using telemedicine for routine care for people with chronic diseases and to provide care for COVID-19 patients with several challenges, again a system similar to other countries during this pandemic [13,14]. About 70–90 percent of the population are able to access the internet and have at least one mobile device [15]. Many people reported that they use internet for seeking health information and have a high level of electronic health literacy [16]. Therefore, use of a teleconsultation service seems to be possible for most people.

To increase the effectiveness of teleconsultation to enhance self-care in cases of COVID-19 was challenging, however meeting the needs of the patients is a predominant task for the health care services. A recent study shows that Thai people have a high awareness of COVID-19 prevention; however, less than half (45.4%) have high levels of knowledge about COVID-19 [17]. The most common sources Thai people use to seek health information is from health care providers and media [18]. Thus, it is essential for physicians and the health care team to provide appropriate and correct answers. Since the information for COVID-19 is always being updated, to recognize and understand the concerns raised by patients needs to guide the team enabling them to prepare and update the information directionally. However, detailed information about the concerns of patients during isolation was lacking. Therefore, this study aimed to determine the topics of concern expressed by individuals with COVID-19 infection receiving care at home via teleconsultation.

## 2. Materials and Methods

We conducted a retrospective study using data from all consenting COVID-19 patients who had consulted the health care team on the online platform during the home isolation period which was between 29 October and 20 December 2021. The study was approved by the Research Ethics Committee of the Faculty of Medicine, Chiang Mai University (8733/2022).

### 2.1. Setting and Participants

This home isolation program was initiated on 29 October 2021, for the provision of health care for patients with confirmed COVID-19 with mild symptoms at first presentation. The patients who were eligible for enrolment onto this program needed to meet the initial criteria. Once registered onto the program, patients had to follow the instruction for self-management at home. The initial criteria and instruction for self-management are detailed in Appendix A and Appendix B During the 10-day home isolation period, patients were able to contact the health care personnel of the home isolation team via text messages (chat) through the online platform with no limitation on frequency of contact. All meals were delivered to the place of isolation of the patient until discharge. The management plan is shown in Figure 1.

### 2.2. Data Collection

Secondary data was obtained anonymously from electronic medical records (for age and gender), and the record of text messages in the teleconsultation service where the patients were able to ask questions freely and express their concerns during the home isolation period. Informed consent was obtained from all patients before the texts were extracted. All text messages were manually reviewed by two professionals (K.P. and N.B.). Questions or statement that were initiated by the patients were included in the analysis as representative of their concerns. The follow up time ended once the patient was hospitalized or made the decision to resign from the home isolation program.

### 2.3. Data Analysis

Descriptive analysis was used to describe participant characteristics including frequency, percentage, mean, and standard deviation. To assess patient concerns, the data were uploaded into NVivo (version 12) qualitative and conducted thematic analysis to code for predetermined themes of interest as well as emergent themes. Two independent researchers (K.P. and N.N.) led the analysis by using the inductive coding approach and discussed preliminary results with other researchers. The identification codes were compared and the similarities and differences for finding a consensus on the emergent themes and sub-themes were reached.

To ensure the reliability of the study, all researchers have worked as physicians in the research field since the home isolation service began. In order to triangulate the results, the researchers presented the results to other healthcare staff who worked on the home isolation program (physicians and nurses) and received their feedback with regard to the agreement. All authors read and contributed to the manuscript.

## 3. Results

Two hundred and thirteen patients were recruited onto the study and agreed their data could be used in the analysis. A small majority of the patients were female (58.69%). The average age was 32.26 ± 16.92 years. A total of 475 questions were generated by 150 patients during the isolation period. Sixty-three patients (29.58%) never asked anything.

From the analysis, the questions generated by patients could be divided into three themes which were: (1) complex care system, (2) uncertainty about self-care and treatment plan with regard to lack of knowledges and skills, and (3) concerns about health recovery and returning safely to the community after post COVID-19 infection (Figure 2). Table 1 shows a summary of themes, subthemes and numbers of quotes per subtheme. The details of each theme were as follows:

### 3.1. Theme: Complex Care System

#### 3.1.1. Several Steps of Registration Process

There were two concerns related to the registration process. The first being about the actual consent form (*n* = 4). In order to enroll on the home isolation program, the patients needed to sign the consent form that was delivered to their home and send an image of the form back to the health care provider. This is to ensure that they were eligible to receive care from Maharaj Nakorn Chiang Mai Hospital and also included an agreement that they will strictly adhere to the instructions for home isolation.

The second concern was about the COVID-19 registration number (*n* = 19). All COVID-19 patients needed to receive a registration number to identify them as an index case. The registration number was necessary on the consent form for identification and was also needed when someone who indicated close contact had to go for a COVID-19 RT-PCR test at the hospital. However, the number was always provided after enrollment by the third party (government sector), so it was completely understandable there were questions about this number at the time of entry to the program.

#### 3.1.2. Complex Monitoring System and Lack of Clarity Re: Received Services

Many concerns related to the complex nature of the monitoring system and services.

-Reporting of vital signs: Problems with the system were reported. (*n* = 28)


*“I could not access the system. The system was down. When should I access again?”*


-Way to communicate with health care providers: The route and available times were asked in order that the patients could contact the team if there were any problems. (*n* = 6)

*“How do we communicate? Do we chat* via *this route?”*

-Other hospital appointments during the isolation: Those who had had contact with COVID-19 cases, were scheduled in advance for their RT-PCR test. Even if they had tested positive, the scheduled appointment was not canceled automatically, so they still received the reminder message. Some patients had health conditions that had scheduled appointments and received phone calls from other hospitals to confirm them. The question was around what they should do about these appointments. (*n* = 4)


*“I receive a phone call from (hospital name). They asked me to pack my bag and go get the treatment at their place. What should I do?”*


-Medical kit box: As some patients received their medication and pulse oximeter, they expressed concern about how long they had to wait to report it. Moreover, the box was provided free of charge, so there was concern as to whether they should return the box and needed to know how. (*n* = 23)


*“My box has not arrived yet; can I use my boyfriend’s pulse oximeter?”*


-Need for extra equipment: Some things were asked for during the isolation including new batteries for the pulse oximeter and alcohol spray. (*n* = 2)-Change of location to receive care: Some patients were not allowed to continue home isolation by the owner of the place they lived. (*n* = 8)


*“I’m sorry that I cannot continue the home isolation. The owner of the dormitory is not okay with me staying here. It is better for me to stay at the hospital with my friend.”*


-Contact other hospital: A patient who’s grandmother was COVID-19 positive and receiving the treatment at another hospital asked for help. (*n* = 1)


*“My grandmother is receiving the treatment at (hospital name). Do you have the contact for that place?*


-Documentation: Certificates were requested by most patients for insurance companies or employers. (*n* = 35)


*“I am a teacher. I need the medical certificate for my school that I was absent because of COVID-19 infection.”*


-Meal services: Meals for patients in home isolation system were distributed by a specific driver under the hospital home isolation system, there were some questions about the meals during home isolation for example at what exact time the meal would come or when would the electronic meal coupon expire. (*n* = 3)

### 3.2. Theme: Uncertainty about Self-Care and Treatment Plan with Regard to Lack of Knowledge and Skills

Many concerns were related to the topics about self-care and treatment for COVID-19. As they had to live alone for 10 days during the home isolation period, self-care was very important as was the remote care to support them.

#### 3.2.1. The Monitoring of Symptoms and Physical Health in Order to Determine the Progression of COVID-19

-Monitoring of vital signs: Questions about ‘How to monitor and how to record the data?’ And ‘How to read pulse oximeter?’ were asked. (*n* = 12)


*“I recorded my vital signs in the system. Did I do it correctly?”*


-Symptom concerns: Covid related symptoms including rash, red eyes, high grade fever, cough, dyspnea, vomiting, anosmia/ageusia, dizziness and fatigue were reported. Management of general health issues about symptoms from other causes were also asked. (*n* = 36)


*“My mom fell down after walking out from the bathroom and had a bump on her head. What should I do?”*


-Indicator of recovery: Many patients were concerned about how to indicate that they had recovered. Some expressed concern that the disease would get worse after it got better. (*n* = 8)


*“If I had already recovered, can the disease still get worse? Or if it is better will I always be better.”*


-Disease progression: Lungs were the organs that caused the highest level of concern amongst patients. Patients asked how would they know if the virus had entered the lungs. Moreover, as the patients were asked to monitor their pulse rate and oxygen saturation, they were sometimes concerned about the numbers. With regard to interventions, chest X-ray was the intervention that the majority of patients believed could help detect the progression of the disease so some requested one. (*n* = 30)


*“How can I know that the virus enters to my lungs? Are my lungs okay now?”*


#### 3.2.2. Several Regimens of Medication Resulting from the in-Advance Medical Box Set and Non-Prescribed Medicine

-Prescribed medication: Even though a leaflet about how to take the medication was provided, the patients still asked for confirmation about what medication, how many, when, and how long should they take it for. In one case, the mother of a child asked about how to store the medication (favipiravir syrup). Most concerns about medication were about their side effect(s), what should be done if they forget, and can they take the anti-viral medication with other medication. Also during the isolation, more medication was requested, as there were some new or prolonged symptom(s). (*n* = 84)


*“I have more cough and phlegm. Can I have more medication for my coughing?”*


-Non-prescribed medication, herbal medicine or supplements: The extract of Andrographis paniculate was the most common medicine that was mentioned. Questions about other herbal medicines like Kaempferia and supplements such as Vitamin C and Zinc were also asked. (*n* = 8)


*“My friend brought me the capsule of Andrographis paniculate extract. But I don’t trust it. Should I take it?”*


#### 3.2.3. Behavioral Health Control While Being Ill with COVID-19

-Diet and drinks: Questions were asked about many kinds of food and drink and if they were fine to be eaten and drunk whilst having COVID-19. The most frequently reported concerns were around alcohol and coffee. (*n* = 22)-Exercise: Some questions about doing exercise during COVID-19 infection was raised by several patients. (*n* = 5)

#### 3.2.4. Mental Health Problems from the Consequences of COVID-19 Infection

-A mother of a boy expressed that she could observe that her son seemed stressed from being in forced isolation. (*n* = 1)


*“I think my son (who is positive and stays in the same house) is not feeling well. He cried and said he was being kept in jail.”*


-Another example is a girl reported that she felt stressed as she is due to travel abroad with her boyfriend soon, but because of being sick with COVID-19 she can’t prepare a lot of documents. (*n* = 1)


*“I will have to travel abroad with my boyfriend for which I need to prepare a lot of documents. I have teeth grinding at night and it was bleeding last night because I grind my teeth stronger than usual from the stress.”*


#### 3.2.5. Infection and Transmission Prevention

-COVID-19 variants: The patients wanted to know about variants of COVID-19. (*n* = 2)


*“Did you test for COVID-19 variant? Is it Delta variant?”*


-Duration of isolation: The first policy stated that the patients need 14 days isolation but later on it was changed to 10 days, so patients were confused with regard to the total number of days required and also the start date of counting. (*n* = 24)


*“How to count the 10 days? Is it from the date we tested positive or what?”*


-Prevention strategies: In cases where patients were staying alone at home, the patient asked for permission not to wear a mask. (*n* = 5)


*“I stay alone at home. Can I not wear the mask? I do not feel well when wearing it. I cannot breathe comfortably.”*


-Trash management: Difficult issues about trash management were reported by some patients. This was usually from those staying in shared space like an apartment or dormitory. (*n* = 4)


*“After spraying the alcohol on the outside bag, where should I place the trash? The owner said they cannot handle the infected bag.”*


#### 3.2.6. Living with Household Members

Some patients expressed concern about the effects of isolation on other family members or pets. Some patients had other family members infected, but were being treated in different places with overlapping time. When they were rejoining the group, the issue about safety when moving back together was raised. During the isolation period, the family members of some patients had a positive test at a later time. (*n* = 22)


*“Will I get infected (superinfection) from my COVID-19 positive friend?”*



*“Now that I am infected with COVID-19, can I touch my cat, or do I need to isolate from it?”*


### 3.3. Theme: Concerns about Health Recovery and Returning Safely to the Community after COVID-19 Infection

During the time they were in the home isolation program, patients had plans for doing something good for their health after they recovered. Several participants asked about re-testing for COVID-19 test after recovery and some concerned related to others.

#### 3.3.1. Further Self-Care

-Vaccination: Most of the patients were in between doses of COVID vaccines and some were about to have the first dose. Other vaccines were also asked about by one patient. (*n* = 17)


*“I received (vaccine name) for my second dose. Which vaccine should I go for next?”*


-Rehabilitation: The patient believed that the lungs were destroyed by the virus. He needed to rehabilitate the lung to normal function. (*n* = 1)


*“If I recover, can I do both aerobic and anaerobic exercise. Which one do you think will decrease the fibrosis of the lungs better after 2 months?”*


-Chance of reinfection: The possibility of reinfection was of concern. (*n* = 4)


*“Is there any possibility that I will be reinfected?”*


-Re-test for COVID-19: Most of the were patients concerned that they still have the virus and can spread it to others. Some asked if a re-test is required. (*n* = 30)


*“When should I test for RT-PCR again? Or will still be positive and not needed?”*


#### 3.3.2. Social and Environment

-House cleaning: There was a belief that disinfectant spray is required when cleaning the house to make sure that it is safe for themselves and other people to stay in the house later. (*n* = 6)


*“If I completed my home isolation, will there be someone to come and spray disinfectant in my house?”*


-Return to work: Concerns about going back to work were high, as patients needed to make sure that their coworkers will be safe and not feel bad about them. (*n* = 20)


*“Today is my 10th day of isolation. Tomorrow I can go out to work, right?”*


## 4. Discussion

The results show three main themes of concerns through questions frequently asked via the teleconsultation service during home isolation. These were the complexity of the care system, uncertainty about the self-care and treatment plan with regard to a lack of knowledge and skills, and concern about health recovery and returning to the community after COVID-19 infection. These are important issues that health care providers needed to address correctly and appropriately. From those three themes the major issues can be divided into two main areas for discussion. The first one is the information about COVID-19 infection that the patients need to know for their self-care and care for others. And the second one is the problems about the support from the telemedicine service.

The information about COVID-19 infection and what the patients need to know should be available. For example, the general information about COVID-19 such as transmission, symptoms, days of isolation, medication, and health promotion and prevention after recovery. In addition, the information about which normal health behaviors are safe when being sick with COVID-19 should be available. This includes exercise, alcohol or caffeine drinking, and diet. Although there was some information available online during the pandemic, some issues were not covered well and some included misinformation and were from unreliable sources [19]. Therefore, the patients needed the facility to ask for direct help from the experts. It has been established in Thailand that health care providers are the most used source of health care knowledge [20]. Evidence also shows that suggestions from health care providers could guide patient behavior such as with regard to receiving vaccination [21], as it has been known that immunization can prevent the risk of severe illness after diagnosis and reduce mortality [22,23,24]. Thus, as health care providers, we needed to prepare all this information to ensure that the patients receive the right information for self-care. However, to diminish the workload, which by necessity increases during any pandemic, the common issues and basic information can be summarized and turned into media content in an easy-to-read format and be given to all patients for self-study. The statements need to be understandable and include low levels of literacy, and when information is updated, the public should be informed without fear of correcting earlier statements if need be [25].

Not surprisingly, in our study, apart from reporting their physical symptoms, some patients reported their mental health problems. There is evidence to show that the risk of mental health disorders such as depression and anxiety increased in patients who had to quarantine [26,27]. It can result from the perceived severity of the disease [27], feelings of loneliness [28], or the perception of forced social isolation [29]. This reminds the health care provider to look for the need for mental health support during the isolation period. It is imperative to include screening for mental health problems, especially depression and anxiety, in the care program.

Issues about the support system for telemedicine service need to be carefully addressed. The patients need an easy to access, user-friendly service from the beginning, including the registration process. It should be easy enough and intuitive so that the patient would not be confused or take much time as there was no assistance on site. Help was available only by phone. During the process of care, telemonitoring or teleconsultation should be implemented with some assistive technologies. Smart health care supportive devices [30], which could connect to and automatically transmit information to the application on the smartphone, can reduce the confusion about where to record and also facilitate real time reporting. Artificial intelligence might help provide general recommendations or help in seeking useful information from approved resources [31].

For the future in which COVID-19 might be moved from being responsible for a pandemic to an endemic disease, believed to have resulted from the availability of COVID-19 vaccinations all over the world and the significant reduction in morbidity and mortality [22], the practice might be slightly different from what has been suggested to the patients during the study. Moreover, clinical manifestations, transmissibility, morbidity, and mortality of COVID-19 from the new variants of the Coronavirus could be potentially changed [32]. Further observations to update recommendations and telemedicine service pattern relating the change is needed.

The strength of this study is that we conducted the review of all the text data from the patients at the time receiving care for COVID-19 treatment at home. The data was collated from actual patients’ phrases not from a self-reporting questionnaire, which could lead to recall bias. This, therefore, could provide the information about what patients really need to know when they get sick during a pandemic. However, the study is not without its limitations. The study did not include the questions that were asked using phone calls, which would have been helpful in grouping responses. Additionally, as the home isolation program only included the patients with mild forms of COVID-19, the concerns may be different in other groups with greater disease severity or with the higher risk of progression. However, this population with a mild form of the disease could have early stages of COVID-19 before progression to more severe forms and could have similar issues of concern.

## 5. Conclusions

The result of this study shows that many aspects of information were enquired about during home isolation that were useful for patient self-care and might guide their future health behavior. To take the findings forward the relevant information needs to be prepared in advance when providing care in this way. The outcomes of our study support the idea that telemedicine was important during the pandemic [33], but the service needs to be user-friendly and health care providers need to understand the system well enough to advise and fix any problems when required. There are still challenges for improvement in the future use of technologies during the pandemic. The findings from this study with regard to teleconsultation have proved useful during the current COVID-19 pandemic situation and also could be implemented in the management of other future infectious diseases in which patient care can be given during home isolation.

## Figures and Tables

**Figure 1 ijerph-19-06591-f001:**
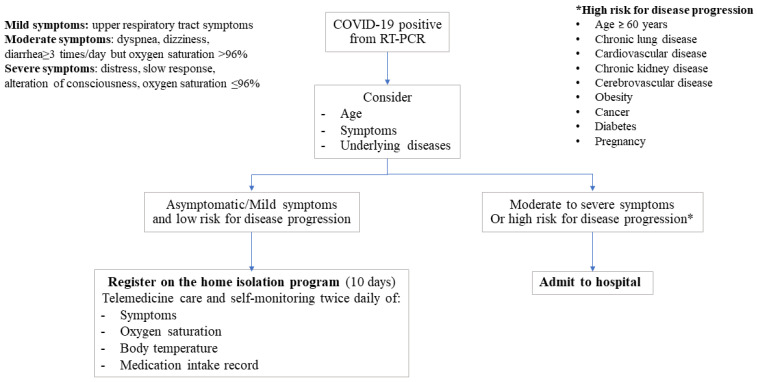
Management plan.

**Figure 2 ijerph-19-06591-f002:**
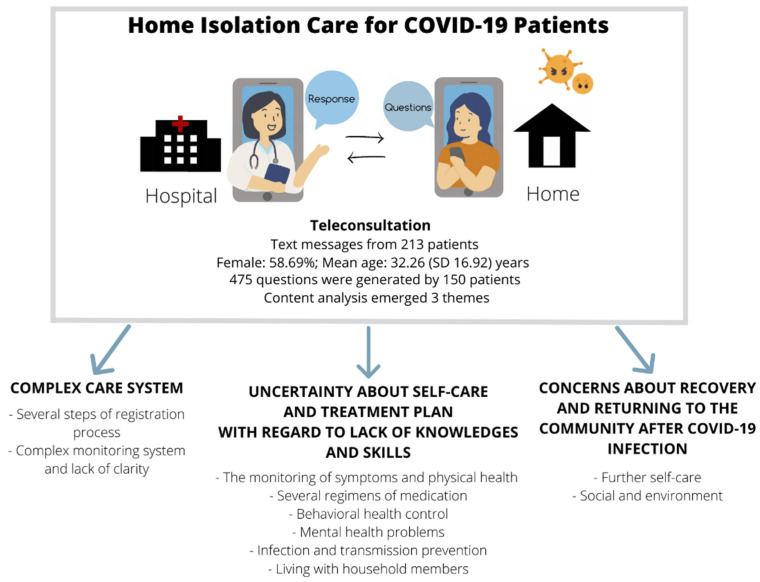
Summary of the results.

**Table 1 ijerph-19-06591-t001:** Themes, subthemes and numbers of questions per subtheme.

Themes	Subthemes	Counts
**1. Complex Care system**	** *Several steps of registration process* **	
	Request for registration no.	4
	Completion of consent form	19
	** *Complex monitoring system and lack of clarity re: received services* **	
	Reporting of vital signs	28
	Ways to communicate with health care providers	6
	Other hospital appointments during the isolation	4
	Medical kit box	23
	Need for extra equipment	2
	Change of location to receive care	8
	Contact other hospital	1
	Documentation	35
	Meal services	3
**2. Uncertainty about self-care and treatment plan with regard to lack of knowledges and skills.**	** *The monitoring of symptoms and physical health in order to determine the progression of COVID-19* **	
	Monitoring of vital signs	12
	Symptom concerns	36
	Indicators of recovery	8
	Disease progression	30
	** *Several regimens of medication resulting from the in-advance medical box set and non-prescribed medicine* **	
	Prescribed medication	84
	Non-prescribed medication, herbal medicine or supplements	8
	** *Behavioral health control while being ill with COVID-19* **	
	Diet and drinks	22
	Exercise	5
	** *Mental health problems from the consequences of COVID-19 infection* **	
	Stress from forced isolation	1
	Traveling abroad after COVID-19 infection	1
	** *Infection and transmission prevention* **	
	COVID-19 variants	2
	Duration of isolation	24
	Prevention strategies	5
	Trash management	4
	** *Living with household members* **	22
**3. Concerns about health recovery and returning safely to the community after COVID-19 infection**	** *Further self-care* **	
	Vaccination	17
	Rehabilitation	1
	Chance of reinfection	4
	Re-test for COVID-19	30
	** *Social and environment* **	
	House cleaning	6
	Return to work	20

## Data Availability

Data are available from Chiang Mai University Ethics Committee (contact via researchmed@cmu.ac.th) for researchers who meet the criteria for access the confidential data.

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
