# Peer review of "Concerns of Home Isolating COVID-19 Patients While Receiving Care via Telemedicine during the Pandemic in the Northern Thailand: A Qualitative Study on Text Messaging"

_ijerph, 2022, doi:10.3390/ijerph19116591_

Round 1

Reviewer 1 Report

Concerns of Home Isolating COVID-19 patients while receiving care via Telemedicine during the pandemic in Northern Thailand: A Qualitative Study on Text Messaging

Thank you for the opportunity to review this paper. The information in the paper is relevant and does not only apply to the COVID-19 pandemic situation but is a telemedicine strategy applicable to many infectious diseases that could be managed using home isolation and teleconsultation that should be perfected.

The objective of the study is however clearly stated in the paper. The main issue with the paper is the background data provided and the context setting which impacts the method, results presentation, and the discussions.

I can see several areas the background data provided could have benefitted from but seems missing. One could have been the background of the study population including that of Thailand and the province where this study was conducted in particular their socio-economic status and health-seeking behavior. The use of telemedicine requires a certain level of education and technology including its use for it to be a viable way for teleconsultation. With that visibility, the reader would understand better the context of the study and the findings. In addition, more details of the available COVID-19 information pack shared with the patients should have been made available to help understand why the patients asked the questions they asked.

The results, discussions, and conclusions are aligned but would require a better context to make the paper more applicable, especially with the availability of COVID-19 vaccinations and the significant reduction in morbidity and mortality due to COVID-19 being experienced all over the world leading to a change in status of covid-19 from pandemic status to epidemic status by the WHO. As indicated earlier, this is a significant and relevant piece of work that should inform telemedicine moving forward.

I will recommend the publication of the paper after making major changes and incorporating some suggestions if the drafters of the manuscript agree.

Author Response

We would like to thank the reviewer for all helpful comments and suggestions.  The manuscript has been revised according to the reviewers’ suggestions and we hope that our response addresses the reviewers’ comments.

Reviewer 2 Report

Dear Authors, 
The work should be considerably improved from my point of view. The introduction should answer the questions: what is the problem, why is the problem important, what have others done to solve the problem and what do you contribute? The introduction needs to be better documented. 
On the other hand, in the material and methods section I would depersonalize the actions, you should not put your role in the design but your actions carried out should be expressed in contributions at the end of the work. 
On the other hand, the conclusion does not show what are the implications of your work. What have you solved? What have you discovered?

Author Response

(The authors gave the same response as above.)

Reviewer 3 Report

The paper presented for a review concerning worries of patients with COVID-19 and possibilities of their reduction using telemedicine contains very current content. Results of these studies may be implemented in a clinical practice and contribute to improvement of patients with COVID-19 life quality. I suggest some refinement in terms of presentation of the study results by reduction of the content of the questions quoted and statements of patients and presentation a broader description with the visualisation of obtained results in a graphic form. Also, the title of table 1 is not compatible with content included in a table – please improve it. After these corrections a paper could be published.    

Author Response

We would like to thank the reviewer for the helpful comments and suggestions.  The manuscript has been revised according to the reviewers’ suggestions and we hope that our response addresses the reviewers’ comments.

Round 2

Reviewer 2 Report

The authors have taken into account the previous considerations and have resolved the suggestions of the various reviewers.